# Cost-Effectiveness Analysis of Photobiomodulation After Third Molar Extraction for Pain Control

**DOI:** 10.3390/ijerph22020159

**Published:** 2025-01-25

**Authors:** Thalita Molinos Campos, Mayra Costanti Vilela Campos, Raquel Agnelli Mesquita-Ferari, Anna Carolina Ratto Tempestini Horliana, Sandra Kalil Bussadori, Cinthya Cosme Gutierrez Duran, Alexandre Padilha, Aldo Brugnera Júnior, Samir Nammour, Ricardo Scarparo Navarro, Kristianne Porta Santos Fernandes, Lara Jansiski Motta

**Affiliations:** 1Postgraduate Program in Biophotonics Medicine, Nove de Julho University (UNINOVE), Rua Vergueiro, 235/249—Liberdade, São Paulo 01504-001, SP, Brazil; thalita.molinos@uni9.pro.br (T.M.C.); mayra.vilela@uni9.edu.br (M.C.V.C.); raquelmesquita@uni9.pro.br (R.A.M.-F.); acrth@uni9.pro.br (A.C.R.T.H.); sandrakalil@uni9.pro.br (S.K.B.); cduran@uni9.pro.br (C.C.G.D.); padilha.alexandre@uni9.pro.br (A.P.); kporta@uni9.pro.br (K.P.S.F.); 2Postgraduate Program in Rehabilitation Sciences, Nove de Julho University (UNINOVE), Rua Vergueiro, 235/249—Liberdade, São Paulo 01504-001, SP, Brazil; 3Postgraduate Program in Medicine, Nove de Julho University (UNINOVE), Rua Vergueiro, 235/249—Liberdade, São Paulo 01504-001, SP, Brazil; 4Education College of the European Master in Oral Laser Applications (EMDOLA), University of Liege, Pl. du Vingt Août 7, 4000 Liège, Belgium; aldo.brugnera@gmail.com; 5IFSC, University of São Paulo, USP, Avenida Trabalhador São-Carlense, n° 400, Parque Arnold Schimidt—CEP, São Carlos 13566-590, SP, Brazil; 6Department of Dental Sciences, Faculty of Medicine, University of Liege, Pl. du Vingt Août 7, 4000 Liège, Belgium; s.namour@uliege.be; 7Bioengineering Postgraduate Program, Scientific and Technological Institute, Universidade Brasil, Rua Carolina Fonseca, 584, São Paulo 08230-030, SP, Brazil; ricardo.navarro@ub.edu.br

**Keywords:** cost-effectiveness, low-level laser, orofacial pain, oral surgery, photobiomodulation

## Abstract

This study aims to evaluate the cost-effectiveness of photobiomodulation applied after third molar extraction. Materials and Methods: To evaluate cost-effectiveness, 15 studies were selected for a systematic review and 8 studies for a meta-analysis to determine the effectiveness of photobiomodulation after surgery. In the present study, as a measure of effectiveness, the pain scale (visual analog scale) was used. The laser value was extracted from the Unified Terminology of Supplementary Health (Brazilian Health System) according to the laser application protocol most common among the clinical trials selected for the meta-analysis. As for drugs, they were determined from those most used among the works included in the meta-analysis and within the protocols established by the Brazilian Ministry of Health. Results: The results of the overall analysis show a significant reduction in pain on the second day after surgery for the experimental group compared to the control (MD, −1.15; 95% CI, −1.73, −0.57). The control group has a lower cost and lower effectiveness, while laser treatment has a higher cost and higher effectiveness. Faced with this situation, the professional must clinically assess whether the cost of USD 34.62 for controlled pain intensity using the laser is worth the extra health benefit. Conclusions: Regarding the cost-effectiveness assessment, the control group has a lower cost and lower effectiveness, while laser treatment has a higher cost and higher effectiveness The decision of which treatment to choose must consider whether the cost of the therapeutic alternative outweighs the clinical gain caused by the treatment. Clinical Relevance: One of the most executed procedures in dentistry is the extraction of third molars. To reduce the negative post-surgical effects, anti-inflammatory drugs are prescribed, which can generate unwanted effects. Photobiomodulation is a technique to modulate inflammation, accelerate tissue repair, and reduce pain and discomfort in different clinical situations.

## 1. Introduction

One of the most executed procedures in dentistry is the extraction of third molars [1], a surgery that causes a lot of discomfort to patients, with pain, swelling, and trismus due to the surgical trauma [2,3,4]. To reduce these negative post-surgical effects, both corticosteroid and non-steroidal anti-inflammatory drugs are prescribed after tooth extraction, but these drugs can generate unwanted effects, such as allergic reactions and gastrointestinal problems, making them often not so safe, which has led to an increase in interest in the search for viable alternative methods, free from adverse effects [5,6,7].

Photobiomodulation (PBM) is a technique to modulate inflammation, capable of reducing pain and discomfort in different clinical situations and accelerating tissue repair [7]. As for its analgesic effect, both for its action on inflammation mediators and for its irradiation in the nervous path, low-intensity laser therapy is capable of penetrating the tissues, influencing both the synthesis and the release and metabolism of substances that are directly involved in analgesia, such as serotonin and acetylcholine, promoting modulation of inflammation locally through its action on mediators, such as histamine and prostaglandins [8,9], stimulating blood flow in the region [10,11].

Regarding its modulating effect on inflammation, the laser promotes an increase in phagocytic activity, increases the diameter of lymphatic vessels, reduces the permeability of blood vessels, and promotes restoration of microcapillary circulation, causing it to normalize the permeability of vascular walls, reducing the edema [11,12]. Low-level laser therapy is also capable of increasing the modulating effect of inflammation by inhibiting IL-6, IL-10, and TNF-α, depending on the parameters used [13].

It is important to know what is present in the scientific literature regarding PBM after third molar extraction. There must be an analysis of the level of evidence and the results of controlled clinical trials so that a correct decision can be made regarding the most appropriate treatment. In the same way, the costs of new therapeutic proposals must also be analyzed, with the aim to support correct planning in the health services [14]. Therefore, this study aims to evaluate the cost-effectiveness of photobiomodulation on pain when applied after third molar extraction.

## 2. Materials and Methods

To determine the effectiveness of photobiomodulation on pain after surgery, we conducted a review and meta-analysis before the economic evaluation. This step has a protocol registered in PROSPERO under the number CRD42020154775. For the initial collection of articles on phototherapy after third molar extraction, we used the international databases PubMed, Web of Science, and Medline. The choice of these databases was due to their relevance to the academic environment and the extent of their scope. The search terms were “third molar” AND “phototherapy”, “laser therapy” AND “third molar”, “photobiomodulation” AND “dental extraction”, and “low-level laser therapy” AND “dental extraction”. The search strategy was carried out using descriptors from Health Science Descriptors (DeCS) and MESH, upon prior consultation.

For the selection of articles, the following inclusion criteria were proposed: a randomized clinical trial, where patients underwent third molar extraction for with subsequent phototherapy; applied intraoral or extraoral phototherapy, where patients were evaluated for pain for at least 3 days after surgery; participants who were treated locally or systemically with pain control drugs; and studies where pain and/or trismus were evaluated. The following exclusion criteria were also proposed: clinical studies that differed from a randomized clinical trial, such as just a clinical trial or case report; and studies where the control group received no intervention or only a placebo. From the articles collected in this first stage, selection was made according to the criteria indicated in the methodological guidelines for the elaboration of a systematic review and meta-analysis of randomized clinical trials by the evaluation of the two reviewers for the systematic review.

According to the methodological guidelines for the elaboration of systematic reviews, for the elaboration of the question and the selection of the studies, the PICO strategy is important, which is an acronym for Patient, Intervention, Comparison, and Outcomes. Thus, the question of the systematic review is as follows: What is the effectiveness of photobiomodulation in pain control in patients after third molar extraction when compared to pain control drugs?

For a critical assessment of the systematic review and meta-analysis, PRISMA was used (Preferred Reporting Items for Systematic Reviews and Meta-Analyses), with the objective of guiding the quality of the reporting of data from the systematic review and meta-analysis. Systematic review articles were rated according to Rob 2.0 for risk of bias.

Of the articles selected in the systematic review, to allow for homogeneity between the works selected for the meta-analysis to be carried out, only the works that evaluated pain after surgery using the VAS (visual analog scale) were selected, and the works that used drugs locally or systemically to control pain and that presented data arranged as mean and standard deviation [15,16,17].

Finally, to assess the quality of evidence for the recommendation strength assessment, the GRADE scale (Grading of Recommendations Assessment, Development and Evaluations) was used, classifying each work as providing a high, moderate, low, or very low quality of evidence.

Cost-effectiveness allows determining whether a treatment should be implemented as a therapeutic measure, being calculated as the difference between the cost of two interventions proposed as a treatment divided by the difference between their consequences (effectiveness). Table 1 presents the cost-effectiveness calculation.

In the present study, as a measure of effectiveness, the pain scale (visual analog scale) was used [19]. The laser value was extracted from the Unified Terminology of Supplementary Health (TUSS—Terminologia Unificada da Saúde Suplementar), according to the laser application protocol most common among the clinical trials selected for the meta-analysis. As for drugs, they were determined from those most used among the works included in the meta-analysis and within the protocols established by the Ministry of Health through the Manual of Specialties in Oral Health. The value of the medicines was obtained from the 2021 maximum price list of medicines by active ingredient of the National Health Surveillance Agency (ANVISA—Agência Nacional de Vigilância Sanitária), according to the average value of the medicines contained in the table, with ICMS at 18%, for medicines in the State of São Paulo [20].

To standardize the economic evaluation, checklists available for reporting investigations, prepared by the scientific community, are used, and one of them is the Consolidated Health Economic Evaluation Reporting Standards (CHEERS), which is specific to the reporting of economic evaluation studies [14].

Meta-analysis by weighted difference between means was performed based on selected dichotomous outcomes. Heterogeneity between studies was calculated using I^2^ statistics, and the analysis adopted the random effects model in the present study. The results were described with their respective 95% confidence interval (95% CI). Calculations were performed using R software (The R Foundation for Statistical Computing, Austria). For all analyses, the significance level was set at *p* < 0.05.

## 3. Results and Discussion

The initial survey using keywords in databases indicated 265 studies. After reading the title and abstract, 24 articles were selected that met the inclusion criteria, as shown in Figure 1. Subsequently, following the analysis of the two trained reviewers, 15 articles were included in the present study, in fact for the systematic review. Nine studies were excluded: 2 studies due to inadequate randomization, 1 study only compared groups in different laser protocols, 4 studies in which the control group received only a placebo without medication, 1 study that compared laser with ozone therapy, and 1 study that used a pulsed light device.

The articles were presented in a table (Table 2), including the authors, country of the authors, year of publication, the number of patients who underwent the study, the type of laser device used, the type of laser application (extra- or intraoral), laser wavelength and its power, pain scale used, means of measuring trismus, frequency of laser application, and medication used in patients.

As observed in the studies included in this work, the most used radiant exposure was 4.0 J/cm^2^ (40.00%). As for the wavelength, it can be observed that it varied a lot between the studies. But the wavelength that was most repeated, being applied in 4 of the 15 studies, was 810 nm. Of the studies analyzed, 4 applied the laser intraorally, another 3 applied it extraorally, and 8 studies applied it in both ways. The frequency of laser application also varied greatly between studies, but it was possible to observe that in 7 studies (46.66%) the laser was applied in a single session after the surgical procedure.

Regarding the medication prescribed to patients in the studies, there was also great variation, but the drugs that were repeated most among the studies were Amoxicillin, which appeared in 10 studies (66.66%), and Ibuprofen, which was prescribed in 4 studies (30.7%). Chlorhexidine Digluconate was recommended as a mouthwash in 9 studies (60.00%). Assessing the risk of bias in the studies, Figure 2 demonstrates the risk of biases individually in each study and for each domain considered in the risk assessment, using the Cochrane collaboration tool.

From the initial search that found 230 articles, 15 were selected for the systematic review previously carried out. Of these, 8 articles met the inclusion criteria for the meta-analysis: they evaluated pain after surgery using the VAS scale and included studies that used medication to control pain in the control group, which presented data arranged as mean and standard deviation.

The articles included in this study underwent an evaluation regarding the quality of the evidence, according to GRADE (Grading of Recommendations Assessment, Development and Evaluation). It was possible to observe that 5 studies presented a moderate degree of quality of evidence, and that the other 3 studies presented a high degree of quality of evidence. Studies with a moderate level of evidence quality were thus attributed mainly due to randomization and blinding, which were not properly described and assured in the methodology.

### 3.1. Effect of Low-Level Laser Therapy on Pain Control After Third Molar Extraction

In total, with the eight studies included in the meta-analysis, this study evaluated 498 patients. According to Figure 3, we can see that 249 patients were in the experimental group, and 249 were in the control group, since, in general, studies evaluate the same number of patients in the laser group and control group, considering one half of the patients in each study group. For this analysis, the mean pain on the visual analog scale (VAS) on the second day after tooth extraction was considered.

The eight studies were pooled for meta-analysis. The forest plot (Figure 3) describes the meta-analysis using a weighted difference of the means between the two comparison groups, that is, the laser group and the control group with drugs to control pain, according to the values of the VAS pain scale. Substantial heterogeneity was observed between the studies (I^2^ = 62%; *p* < 0.01), and for this reason the random effects model was chosen. The results of the overall analysis evidence a significant reduction in pain on the second day after surgery for the experimental group compared to the control (MD, −1.15; 95% CI, −1.73, −0.57).

Regarding the analysis performed on the seventh day after tooth extraction, only six studies were considered; those that did not present the mean and standard deviation for the evaluation of pain on the seventh day after surgery were excluded.

The forest plot(Figure 3) also describes the meta-analysis through a weighted difference of means between the two comparison groups, according to the VAS pain scale values. Substantial heterogeneity was observed between studies (I^2^ = 65%; *p* < 0.01). We also opted for the random effects model. The results of this analysis do not show a statistically significant reduction in pain on the seventh day after surgery for the laser group compared to the control group with pain control drugs (MD, −0.30; 95% CI, −0.69, 0.08).

### 3.2. Cost-Effectiveness Analysis

For the cost-effectiveness analysis, the pain level of the visual analog scale was considered effective.

Regarding the cost-effectiveness analysis, the cost of a single laser therapy session was sourced from the Unified Terminology of Supplementary Health (TUSS), where it is listed as USD 39.60 per session under the code 31602215—LASER—PER SESSION. According to the protocols followed in the studies included in the meta-analysis, laser therapy was most commonly applied immediately after surgery, with subsequent sessions on the 2nd and 4th days (three sessions in total).

For the cost analysis, the laser therapy session cost was initially calculated in Brazilian reais and subsequently converted into US dollars based on the exchange rate at the time. While the conversion was made using the applicable exchange rate from the Central Bank of Brazil, it is important to note that the core findings of our cost-effectiveness analysis remain unaffected by the changes in exchange rates. The adjusted calculations reflect the cost of laser therapy and pain medications, with the relative cost-effectiveness between treatments remaining consistent throughout the analysis.

According to the recommendation of the Manual of Specialties in Oral Health, the following drug protocol was established for the cost-effectiveness analysis: Amoxicillin 500 mg (1 pill every 8 h for 7 days = 21 pills), Diclofenac Sodium 50 mg (1 pill every 8 h for 5 days = 15 pills), and Paracetamol 500 mg (1 pill every 6 h for 3 days = 12 pills). Drug prices were obtained from the list of maximum drug prices per active ingredient published by the Brazilian National Health Surveillance Agency (ANVISA—Agência Nacional de Vigilância Sanitária) for 2024 (Table 3 and Table 4). The prices were determined by calculating the average cost of the drugs listed in the table, considering the ICMS rate of 18% for pharmaceuticals in the State of São Paulo. For this average, the prices of boxes containing 20 pills of Diclofenac Sodium 50 mg, boxes containing 21 pills of Amoxicillin 500 mg, and boxes containing 20 pills of Paracetamol 500 mg were used.

To calculate the effectiveness, the mean difference in pain reporting (VAS) in the laser group and the control group of the studies included in the meta-analysis was considered.

Neither of the two treatments is dominant over the other. When a procedure has lower effectiveness and lower cost or vice versa (A or I), it is necessary to perform the ICER (Incremental Cost-Effectiveness Ratio). In this case, whoever makes the decision to choose the treatment needs to assess whether the additional cost of the therapeutic alternative is compensated for by the clinical gain caused by the treatment. The latter was the case for the two treatments evaluated in this study. The control group has a lower cost and lower effectiveness, while laser treatment has a higher cost and higher effectiveness. Faced with this situation, the professional must clinically assess whether the cost of USD 34.62 for controlled pain intensity for the laser is worth the extra health benefit (Table 5).

According to what can be observed in this systematic review and meta-analysis, laser therapy is effective in controlling pain after third molar extraction.

In the results, when describing the laser application protocols, it is observed that they are very divergent, with different wavelengths and number of laser therapy sessions. Oliveira et al. [15] when performing a systematic review and meta-analysis with 374 patients on low-level laser treatment applied after third molar extraction, reported that the most effective wavelengths were between 600 nm and 900 nm. Infrared spectrum lasers (between 780 nm and 904 nm) demonstrated an excellent level of tissue penetration, which may explain why equipment that is within this wavelength range ends up being more effective.

As for the application of the laser being intraoral or extraoral, all studies indicated the mode of application. In the systematic review and meta-analysis [15], despite also observing variation between studies, the intraoral mode or a combination of the two modes of application showed more benefits for edema and trismus. Refs. [16,17,30,31] state that extraoral application has greater benefit for edema and trismus. Refs. [18,19,32], in a clinical study where they divided the laser and placebo groups into intraoral and extraoral, comparing the two groups, found a statistically significant difference in the intraoral group, while the extraoral group had no statistical difference when compared to the placebo group.

Also, regarding laser dosimetric parameters, laser power also varied greatly among the studies included in the systematic review. But in general, the scientific literature suggests that the power with the highest degree of effectiveness is between 10 mW and 500 mW [18,19,31,33].

Some studies do not even provide all dosimetric parameters. This, in addition to hampering a more accurate assessment of the effects of laser therapy and its benefits, also makes it more difficult to standardize a protocol for this procedure, which is important for practice guidance and clinical applicability, and the introduction of the procedure in health services. According to what was possible to observe in this study, we can say that photobiomodulation applied after the third molar extraction is a very common and interesting subject for dentistry.

Assessing the risk of bias in the studies, some studies were evaluated as uncertain in randomization and blinding, as they did not describe and ensure how this was done, even stating that both were performed. It is worth emphasizing the importance of better description of protocol and design in clinical trials, so that, in the face of these analyses, they can be better evaluated.

The meta-analysis showed a statistically significant difference, showing a reduction in pain through laser application on the second day after third molar extraction. This was also found in the meta-analysis carried out by Oliveira et al. [16], though not that by Domah et al. [18], who, in a meta-analysis study with 1064 patients, found no statistically significant difference in pain reduction after third molar extraction, comparing the use of the laser to a placebo.

Regarding the costs of laser therapy after third molar extraction, neither treatment dominates the other. The control group, where patients were treated only with pain medication, had lower costs and lower effectiveness. The laser group, on the other hand, presented higher effectiveness and higher costs. In this case, both treatments point to the need to perform the incremental cost-effectiveness ratio, to know the additional cost of the treatment alternative in view of the clinical gain thereby. Therefore, the cost of a laser is USD 34.62 per controlled pain intensity, and the professional must judge whether the cost is compensated for by the extra health benefit. Perhaps it is plausible, in addition to thinking about the direct health benefit, to also consider the patient’s socioeconomic conditions.

## 4. Conclusions

Despite the variability of laser dosimetric protocols and parameters, photobiomodulation is effective for pain control after third molar extraction, when compared to the placebo group using pain control drugs.

Of the studies analyzed in this systematic review, although quite varied, the most applied wavelength was 810 nm. Its most common application was intra- and extraorally combined, and 46.66% of the studies applied it in a single session, right after the surgical procedure.

Regarding cost-effectiveness, as no group was superior to the other, the choice of the therapeutic alternative should consider whether the cost is compensated for by the clinical gain caused by the treatment, with the cost of laser therapy being USD 34.62 per controlled pain intensity.

## Figures and Tables

**Figure 1 ijerph-22-00159-f001:**
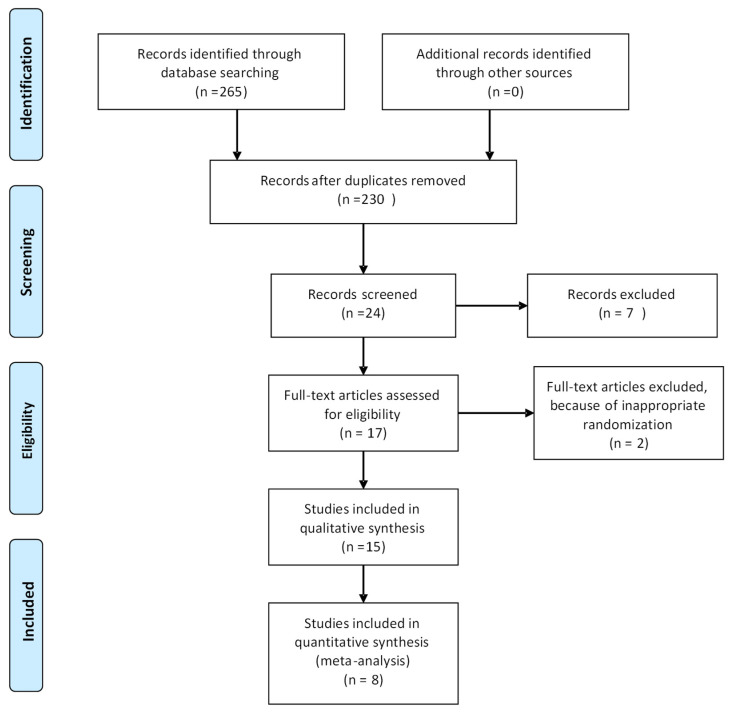
Flow chart of article selection.

**Figure 2 ijerph-22-00159-f002:**
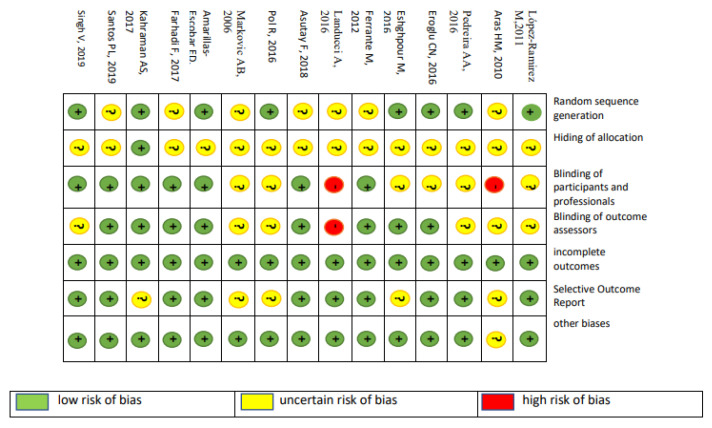
Individual risk of bias of the fifteen studies selected for the systematic review, for each domain of risk of bias assessment of randomized clinical trials, using the Cochrane collaboration tool.

**Figure 3 ijerph-22-00159-f003:**
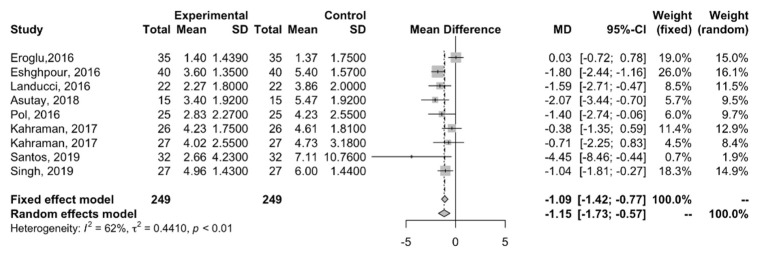
Forest plot with data from meta-analysis studies with assessment on the second day after tooth extraction.

**Table 1 ijerph-22-00159-t001:** Calculation model of cost-effectiveness.

Type of Analysis	Cost Measure	Monetary Value	Result Measure
Cost-effectiveness	Monetary value	Pain	USD/acquired outcome measure [(C1 − C2)/(Q1 − Q2)] C1—laser group cost C2—control group cost Q1—laser group outcome Q2—control group outcome

Source: Adapted from Drummond et al. (2015) [18].

**Table 2 ijerph-22-00159-t002:** Articles included in the systematic review.

Authors	Country	Year	Number of Patients	Type of Laser	Type of Application	Radiant Exposure (J/cm^2^)	Laser Wavelength (nm)	Power (mW)	Pain Scale	Measuring Trismus	Laser Periodicity	Medication
López-Ramirez et al. [6]	Spain	2011	20	GaAlAs	Intraoral	4	810	500	EVA (0–10)	mm of opening	1× after surgery	Amoxicillin, Dipyrone, Ibuprofen, and Chlorhexidine,
Aras et al. [16]	Turkey	2010	48	GaAlAs	Intraoral and extraoral	4	808	100	EVA (0-10)	mm of opening	1× after surgery	Amoxicillin, Paracetamol, and Chlorhexidine
Pedreira et al. [21]	Brazil	2016	24	GaAlAs	Extraoral	2	660–690/790/830	630	EVA (0–10)	mm of opening	1× after surgery and 3× within a week	Dexamethasone and Dipyrone
Eroglu et al. [18]	Turkey	2016	35	Diodo	Extraoral	4	940	500	EVA (0–10)	mm of opening	1× after surgery	Flurbiprofen, Benzydamine, and Chlorhexidine
Eshghpour et al. [22]	Iran	2016	40	InGaAlAs	Extra- and intraoral	--	660/810	200	EVA (0–10)	--	Intraoral—1× after surgery/extraoral— on the 2nd and 4th day after surgery	Amoxicillin, Ibuprofen, and Chlorhexidine
Ferrante et al. [7]	Italy	2012	30	Diodo	Extra- and intraoral	--	980	300	EVA (0–10)	mm of opening	After surgery and after 24 h	Amoxicillin and ketoprofen
Landucci et al. [23]	Brazil	2016	22	GaAlAs	Extra- and intraoral	7.5	780	100	EVA (0–10)	mm of opening	1× after surgery	Amoxicillin, Ibuprofen, and Chlorhexidine
Asutay et al. [24]	Turkey	2018	45	GaAlAs	Extraoral	4	810	300	EVA (0–10)	mm of opening	1× after surgery	Amoxicillin with Clavulanic Acid, Paracetamol, and Chlorhexidine
Pol et al. [25]	Italy	2016	25	GaAs	Intraoral	--	904 and 910	500	EVA (0–100)	--	Intraoral after extraction, then two more sessions at 24 and 48 h	Antibiotic prophylaxis (2 g) and Ibuprofen
Markovic et al. [9]	Serbia and Montenegro	2006	90	GaAlAs	Intraoral	4	637	500	EVA (mm)	--	1× after surgery	Diclofenac
Amarillas-Escobar et al. [26]	Mexico	2010	30	Diodo	Extra- and intraoral	4	810	100	EVA (0–10)	mm of opening	1× Intraoral after surgery. Extraoral after 24, 48 and 72 h.	Dexamethasone, Amoxicillin, Paracetamol, and Ketorolac Trometamol
Farhadi et al. [27]	Iran	2017	48	---	Extra-and intraoral	5	550	100	EVA (0–10)	mm of opening	1× after surgery	Amoxicillin and Ibuprofen
Kahraman et al. [17]	Turkey	2017	60	GaAlAs	Extra- and intraoral	3	830	100	EVA (0–10)	--	Extra- and intraoral before and after surgery	Amoxicillin and Chlorhexidine
Santos et al. [28]	Brazil	2019	32	---	Intraoral	52.5	780	70	EVA (mm)	--	1× after surgery, in 24 h, 48 h, and 72 h.	Medications preferred by the patient. Only Chlorhexidine was prescribed.
Singh et al. [29]	India	2019	25	GaAsAl	Extra- and intraoral	---	830	30	EVA (cm)	mm of opening	1× after surgery, then on the 2nd, 4th, and 7th day	Amoxicillin with Clavulanic Acid, Diclofenac, and Chlorhexidine

**Table 3 ijerph-22-00159-t003:** Survey of costs and value of procedures, G1—photobiomodulation and medication.

Direct Costs, G1			
Medicines/Intervention	Values Medicines/Intervention	Amount Used	Value in the Procedure
Medication: Amoxicillin 500 mg	USD 0.37	21 pills	USD 7.80
Medication: Diclofenac Sodium 50 mg	USD 0.28	15 pills	USD 4.20
Medication: Paracetamol 500 mg	USD 0.20	12 pills	USD 2.40
Laser session	USD 39.60	3 sessions	USD 118.80
Total amount per procedure			USD 133.20

**Table 4 ijerph-22-00159-t004:** Survey of costs and value of procedures, G2—pain control drugs.

Direct Costs, G2			
Medicines/Intervention	Values Medicines/Intervention	Amount Used	Value in the Procedure
Medication: Amoxicillin 500 mg	USD 0.37	21 pills	USD 7.80
Medication: Diclofenac Sodium 50 mg	USD 0.28	15 pills	USD 4.20
Medication: Paracetamol 500 mg	USD 0.20	12 pills	USD 2.40
Total Amount per procedure			USD 14.40

**Table 5 ijerph-22-00159-t005:** Cost-effectiveness analysis—outcome pain.

Type of Analysis	Cost Measure	Outcome Measure	Result Measure
Cost-effectiveness	Monetary value	Pain control Clinical success	USD/acquired outcome measure [C1 − C2)/(Q1 − Q2)]
incremental cost			USD = (14.40 − 133.20)/ (2.62 − 3.32) = 169.70 USD169.70
**Alternative**	**Cost**	**Effectiveness**	**Cost-effectiveness**
Laser group	USD 133.20	3.32 intensity of controlled pain	USD 40.12/by controlled pain intensity
Placebo group	USD 14.40	2.62 intensity of controlled pain	USD 5.50/by controlled pain intensity
Difference per unit of controlled pain intensity			(USD 40.12 − 5.50)/by controlled pain intensity USD 34.62/by controlled pain intensity

## Data Availability

https://www.crd.york.ac.uk/prospero/display_record.php?ID=CRD42020154775.

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
