# Peer review of "Cost-Effectiveness Analysis of Photobiomodulation After Third Molar Extraction for Pain Control"

_ijerph, 2025, doi:10.3390/ijerph22020159_

Round 1

Reviewer 1 Report

Comments and Suggestions for Authors

The manuscript refers to a systematic review concerning "Cost-effectiveness analysis of photobiomodulation after third molar extraction for pain control".

Overall the subject is of IJERPH readers' interest.

Authors have conducted a systematic review respecting all actual SR rules: PROSPERO registration, Prisma Statement requirements, inclusion criteria of gold standard publications (only RCTs) and a meta-analysis evidencing statistical power.

The only drawback of the manuscript presented is the excessive number of authors, which is not justified for a manuscript such as the one submitted. Systematic reviews should be conducted by three researchers, two responsible for the initial searches and consensus meetings, including a third to resolve possible doubts. The manuscript in question lists 12 authors, which is far beyond what is acceptable when referring to responsible scientific production. 

Authors proposed to analyze the cost-effectiveness of photobiomodulation after third molar extraction surgeries by a systematic review and meta-analysis. Methodology: considering the included studies (inclusion criteria - only RCTs) it could be considered as gold standard to answer other PICO questions, but cost-effectiveness was not treated by the included studies original authors and the submitted paper treat this subject by subjective analysis/"narrative review" literature.
 Authors tried to bring some originality to the presented manuscript by adding a subjective analysis regarding cost-effectiveness using literature data beyond included studies of the systematic review.
Considering recent published articles: doi: 10.3390/app14073049 ; doi: 10.1016/
j.jebdp.2024.101983 ; doi: 10.1016/j.joms.2023.05.007 and others, data content brings no novelty.

 LINES 344-350: present the same conclusion as previous published systematic reviews.  - LINES 351-354: no conclusive point for clinical decision was described. Authors conducted the systematic review methodology appropriately. The main concern is why they included 15 studies in systematic review and only 8 in the presented meta-analysis? (LINES 29-30)
Authors Appropriately use tables and figures.   

Author Response

Dear Reviewer,

We would like to express our sincere gratitude for your thoughtful and constructive feedback. We appreciate the time and effort you have dedicated to reviewing our manuscript, and we are grateful for your valuable insights that have significantly contributed to improving the quality of our work.

We have carefully addressed all the comments and suggestions provided and have made the necessary revisions to ensure that the manuscript meets the highest standards. Below, we outline our responses to each of the comments, along with the corresponding modifications made to the manuscript.

Comment 1:
The manuscript refers to a systematic review concerning "Cost-effectiveness analysis of photobiomodulation after third molar extraction for pain control".
Overall the subject is of IJERPH readers' interest.
Authors have conducted a systematic review respecting all actual SR rules: PROSPERO registration, Prisma Statement requirements, inclusion criteria of gold standard publications (only RCTs) and a meta-analysis evidencing statistical power.
The only drawback of the manuscript presented is the excessive number of authors, which is not justified for a manuscript such as the one submitted. Systematic reviews should be conducted by three researchers, two responsible for the initial searches and consensus meetings, including a third to resolve possible doubts. The manuscript in question lists 12 authors, which is far beyond what is acceptable when referring to responsible scientific production.

Response 1:
Thank you for your positive comments regarding the relevance of the manuscript to the IJERPH readership. We also appreciate your feedback on the structure and methodology of the systematic review.
Regarding the number of authors, we would like to clarify that in addition to conducting the systematic review, we also performed the meta-analysis and the cost-effectiveness analysis. Given the use of multiple methodologies in the study, we required the support and expertise of a larger team of researchers to ensure the robustness and accuracy of the analyses. Each author contributed significantly to specific aspects of the study, including the systematic review, meta-analysis, and cost analysis. Thus, the involvement of a diverse group of researchers was essential for the comprehensive nature of this work.
We hope this explanation addresses your concern, and we are happy to provide further clarification if needed.

Comment 2:
Authors proposed to analyze the cost-effectiveness of photobiomodulation after third molar extraction surgeries by a systematic review and meta-analysis. Methodology: considering the included studies (inclusion criteria - only RCTs) it could be considered as gold standard to answer other PICO questions, but cost-effectiveness was not treated by the included studies' original authors, and the submitted paper treats this subject by subjective analysis/"narrative review" literature.

Response 2:
Thank you for your thoughtful comments. We would like to clarify that our work was conducted in two distinct phases. In the first phase, we focused on conducting the systematic review and meta-analysis with the objective of defining an effectiveness measure for photobiomodulation following third molar extractions. This phase strictly included randomized controlled trials (RCTs) that evaluated the effectiveness of laser treatment, and no studies analyzing costs were considered at this stage.
In the second phase, we used the effectiveness measure derived from the systematic review and meta-analysis to guide the cost-effectiveness analysis. Since the studies included in the systematic review did not provide cost data, we applied a narrative review methodology to address the cost-effectiveness aspect, based on available literature that could inform this analysis. Thus, the information extracted from the systematic review and meta-analysis was focused on the effectiveness of the treatment, which was then applied to the cost analysis in the subsequent phase.
We hope this explanation clarifies the methodological approach we adopted in our study.

Comment 3:
Authors tried to bring some originality to the presented manuscript by adding a subjective analysis regarding cost-effectiveness using literature data beyond included studies of the systematic review.
Considering recent published articles: doi: 10.3390/app14073049; doi: 10.1016/j.jebdp.2024.101983; doi: 10.1016/j.joms.2023.05.007 and others, data content brings no novelty.
LINES 344-350: present the same conclusion as previous published systematic reviews.
LINES 351-354: no conclusive point for clinical decision was described.
Authors conducted the systematic review methodology appropriately. The main concern is why they included 15 studies in the systematic review and only 8 in the presented meta-analysis? (LINES 29-30)
Authors appropriately use tables and figures.

Response 3:
Thank you for your valuable comments. We would like to respectfully disagree with the assessment that our manuscript lacks originality. While we acknowledge that our findings are similar to those in the articles you referenced, our primary aim was not to assess the effectiveness of photobiomodulation per se, as addressed by the studies you mentioned. Rather, our goal was to conduct an economic evaluation of photobiomodulation, specifically focusing on the cost-effectiveness of laser therapy after third molar extraction. As you rightly pointed out, the studies you referenced do not present this type of economic analysis, nor does any other literature to the best of our knowledge. Therefore, our study introduces new and relevant information that adds value to the existing body of research by addressing this important economic aspect.
Regarding the inclusion of 8 studies in the meta-analysis out of the 15 included in the systematic review, we would like to clarify that all 15 studies met the inclusion criteria for the systematic review. However, as described in the methodology section, we included only the studies that scored the highest on the GRADE scale and the ROB tool in the meta-analysis. This approach was taken to minimize bias and ensure that the studies included in the meta-analysis were of the highest quality, which is crucial for accurately estimating the effectiveness measure.
Lastly, regarding the concern that our conclusions are not conclusive for clinical decision-making, we would like to emphasize that our study provides a clear conclusion that can guide clinicians in making informed decisions. The information we provide, specifically regarding the cost-effectiveness of laser therapy, offers valuable insights that can help clinicians assess the trade-off between treatment costs and clinical benefits in their practice. We believe that this information contributes to better decision-making and planning for healthcare management.
We hope this explanation addresses your concerns, and we appreciate your thoughtful review.

Reviewer 2 Report

Comments and Suggestions for Authors

This article focuses on an interesting contemporary topic. The extraction of third molars is still a widely performed surgical procedure. Despite its long history and many studies, there is still no single postoperative treatment protocol proven to be the most effective. Photoinhibition is a modern method for modulating the postsurgical healing process. The effectiveness of laser therapy is well established. However, cost-effectiveness analysis in this context is a relatively new field and is particularly important for everyday practitioners.

The manuscript is well-structured, containing an Abstract, Introduction, Materials and Methods, Results and Discussion, and Conclusions. The methods used are clearly presented, and the cited articles are well-organized in a table. Possible biases are acknowledged. A notable weak point is that the most recent article cited for the analysis is from 2019, with no references from the last three years.

The results and discussion are well-supported by the presented data. The finding that laser therapy after third molar extraction has a positive effect is already well known. Similarly, the observation that costs are higher for the laser therapy group compared to the control group receiving only pain medication is not new. However, what is novel and interesting is the precise calculation of the cost associated with controlled pain intensity. It should be emphasized that this data is specific to Brazil and reflects conditions at the time the article was written. While this is a logical conclusion, the authors could further highlight this point.

The main concern is the outdated nature of the cited and analyzed articles. Most of the references are older than 2019, with only one from 2024 and two from 2021. Additionally, the cost analysis relies on drug prices and USD exchange rates from 2021. The overall impression is that this article was largely constructed and written several years ago (likely around 2022) and has only recently been revised.

The article will be of interest to everyday practitioners, oral surgeons, and specialists in public healthcare and healthcare management. Its structure is appropriate, and the results are not merely a repetition of already known information.

Recommended Revisions:

  1. Conduct a new literature search to include the latest articles relevant to the study.

  2. Perform updated calculations using the latest USD exchange rates and drug/laser prices to make the study more reflective of the current social and economic realities.

By addressing these points, the study can be updated and made more relevant to present-day circumstances.

Author Response

Dear Reviewer,

We would like to express our sincere gratitude for your thoughtful and constructive feedback. We appreciate the time and effort you have dedicated to reviewing our manuscript, and we are grateful for your valuable insights that have significantly contributed to improving the quality of our work.

We have carefully addressed all the comments and suggestions provided and have made the necessary revisions to ensure that the manuscript meets the highest standards. Below, we outline our responses to each of the comments, along with the corresponding modifications made to the manuscript.

Comment 1:
This article focuses on an interesting contemporary topic. The extraction of third molars is still a widely performed surgical procedure. Despite its long history and many studies, there is still no single postoperative treatment protocol proven to be the most effective. Photoinhibition is a modern method for modulating the postsurgical healing process. The effectiveness of laser therapy is well established. However, cost-effectiveness analysis in this context is a relatively new field and is particularly important for everyday practitioners.
The manuscript is well-structured, containing an Abstract, Introduction, Materials and Methods, Results and Discussion, and Conclusions. The methods used are clearly presented, and the cited articles are well-organized in a table. Possible biases are acknowledged. A notable weak point is that the most recent article cited for the analysis is from 2019, with no references from the last three years.

Response 1:
Thank you for your thoughtful feedback. We appreciate your comment regarding the references used in the manuscript. We agree that the most recent cited article is from 2019, and we recognize the importance of including the latest studies. In response, we conducted a new literature search to ensure that no relevant RCTs had been published after our initial search that would meet the inclusion criteria for the systematic review and meta-analysis. We confirmed that while there are more recent studies, they do not fulfill the inclusion criteria for this analysis.
However, we value the importance of updating the literature and discussion. Therefore, we have made sure to include more recent references related to laser therapy and third molar extraction in both the Introduction and Discussion sections to provide a more current bibliographic foundation and ensure a more up-to-date context for our findings.
The following references have been added to the manuscript:

  1. Pereira DA, Bonatto MS, Soares EC Jr, Mendes PGJ, Pessoa RSE, de Oliveira GJPL. Photobiomodulation With Infrared and Dual-Wavelength Laser Induces Similar Repair and Control of Inflammation After Third Molar Extraction: A Double-Blinded Split-Mouth Randomized Controlled Trial. J Oral Maxillofac Surg. 2024 Nov 20:S0278-2391(24)00971-6. doi: 10.1016/j.joms.2024.11.009. Epub ahead of print. PMID: 39645230.
  2. Giansiracusa A, Parrini S, Baldini N, Bartali E, Chisci G. The Effect of Photobiomodulation on Third Molar Wound Recovery: A Systematic Review with Meta-Analysis. J Clin Med. 2024 Sep 12;13(18):5402. doi: 10.3390/jcm13185402. PMID: 39336889; PMCID: PMC11432466.
  3. Camolesi GCV, Silva FFVE, Aulestia-Viera PV, Marichalar-Mendía X, Gándara-Vila P, Pérez-Sayáns M. IS THE PHOTOBIOMODULATION THERAPY EFFECTIVE IN CONTROLLING POST-SURGICAL SIDE EFFECTS AFTER THE EXTRACTION OF MANDIBULAR THIRD MOLARS? A SYSTEMATIC REVIEW AND META-ANALYSIS. J Evid Based Dent Pract. 2024 Jun;24(2):101983. doi: 10.1016/j.jebdp.2024.101983. Epub 2024 Feb 28. PMID: 38821660.
  4. Yazdani J, Eslami H, Ghavimi M, Eslami M. Adjunctive Effect of Photobiomodulation Therapy with Nd:YAG Laser in the Treatment of Inferior Alveolar Nerve Paresthesia. Photobiomodul Photomed Laser Surg. 2024 Mar;42(3):208-214. doi: 10.1089/photob.2023.0159. PMID: 38512321
  5. Lu Z, Bingquan H, Jun T, Fei G. Effectiveness of concentrated growth factor and laser therapy on wound healing, inferior alveolar nerve injury and periodontal bone defects post-mandibular impacted wisdom tooth extraction: A randomized clinical trial. Int Wound J. 2024 Jan;21(1):e14651. doi: 10.1111/iwj.14651. PMID: 38272792; PMCID: PMC10789919.

We hope this revision addresses your concern and enhances the manuscript's relevance to contemporary research.

Comment 2:
The results and discussion are well-supported by the presented data. The finding that laser therapy after third molar extraction has a positive effect is already well known. Similarly, the observation that costs are higher for the laser therapy group compared to the control group receiving only pain medication is not new. However, what is novel and interesting is the precise calculation of the cost associated with controlled pain intensity. It should be emphasized that this data is specific to Brazil and reflects conditions at the time the article was written. While this is a logical conclusion, the authors could further highlight this point.
The main concern is the outdated nature of the cited and analyzed articles. Most of the references are older than 2019, with only one from 2024 and two from 2021. Additionally, the cost analysis relies on drug prices and USD exchange rates from 2021. The overall impression is that this article was largely constructed and written several years ago (likely around 2022) and has only recently been revised.
The article will be of interest to everyday practitioners, oral surgeons, and specialists in public healthcare and healthcare management. Its structure is appropriate, and the results are not merely a repetition of already known information.

Response 2:
Thank you for your valuable feedback. We appreciate your recognition of the novelty and precision of our cost analysis related to controlled pain intensity.
In response to your comment, we would like to highlight that we have revised the cost calculations based on Brazilian costs for 2024 and also updated the USD exchange rate to reflect international rates for 2024. This ensures that the cost analysis is aligned with current economic conditions, making the results more relevant and accurate.
We understand the importance of emphasizing the specific context of Brazil, and we have included this update in the manuscript to clarify that the data reflects the most current available figures for drug prices and exchange rates in 2024.
We hope this revision addresses your concern regarding the timeliness of the data and further strengthens the manuscript.

We hope that the revisions we have made, based on your feedback, have enhanced the manuscript's clarity and robustness. We have made every effort to address your concerns and believe that the updated version now reflects the most current data, methodologies, and analysis, which will provide valuable insights to the readers of IJERPH.

Thank you again for your constructive comments.